# The Use of Biosilica to Increase the Compressive Strength of Cement Mortar: The Effect of the Mixing Method

**DOI:** 10.3390/ma16165516

**Published:** 2023-08-08

**Authors:** Nelli G. Muradyan, Avetik A. Arzumanyan, Marine A. Kalantaryan, Yeghiazar V. Vardanyan, Mkrtich Yeranosyan, Malgorzata Ulewicz, David Laroze, Manuk G. Barseghyan

**Affiliations:** 1Faculty of Construction, National University of Architecture and Construction of Armenia, 105 Teryan Street, Yerevan 0009, Armenia; 2Innovation Center for Nanoscience and Technologies, A.B. Nalbandyan Institute of Chemical Physics NAS RA, 5/2 P. Sevak Street, Yerevan 0014, Armenia; myeranos@ysu.am; 3Faculty of Civil Engineering, Czestochowa University of Technology, Dabrowskiego 69 Street, PL 42-201 Czestochowa, Poland; 4Instituto de Alta Investigación, Universidad de Tarapacá, Casilla 7D, Arica 1000000, Chile; dlarozen@academicos.uta.cl

**Keywords:** biosilica, compressive strength, water absorption, cement mortar, mixing method

## Abstract

In this work, the effect of biosilica concentration and two different mixing methods with Portland cement on the compressive strength of cement-based mortars were investigated. The following values of the biosilica concentration of cement weight were investigated։ 2.5, 5, 7.5, and 10 wt.%. The mortar was prepared using the following two biosilica mixing methods: First, biosilica was mixed with cement and appropriate samples were prepared. For the other mixing method, samples were prepared by dissolving biosilica in water using a magnetic stirrer. Compressive tests were carried out on an automatic compression machine with a loading rate of 2.4 kN/s at the age of 7 and 28 days. It is shown that, for all cases, the compressive strength has the maximum value of 10% biosilica concentration. In particular, in the case of the first mixing method, the compressive strength of the specimen over 7 days of curing increased by 30.5%, and by 36.5% for a curing period of 28 days. In the case of the second mixing method, the compressive strength of the specimen over 7 days of curing increased by 23.4%, and by 47.3% for a curing period of 28 days. Additionally, using the first and second mixing methods, the water absorption parameters were reduced by 22% and 34%, respectively. Finally, it is worth noting that the obtained results were intend to provide valuable insights into optimizing biosilica incorporation in cement mortar. With the aim of contributing to the advancement of construction materials, this research delves into the intriguing application of biosilica in cement mortar, emphasizing the significant impact of mixing techniques on the resultant compressive strength.

## 1. Introduction

Currently, cementitious materials are the most widely used construction materials. However, they have low tensile strength and are highly susceptible to cracking. To address this issue, several research efforts have focused on enhancing the cement structure with micro- or nano-level reinforcements. The cement mortars were produced using microsilica [1,2,3,4,5,6,7] as well as nanosilica [8,9,10,11,12,13,14,15,16,17,18,19,20,21]. Both materials were also used together to modify the properties of the mortars, which improved the mechanical properties of the tested mortars [2,3,4,7]. The results of the researched cement composites with nanosilica confirmed its acceleration action on C_3_S hydration and formation gel C-S-H, and the modification of composite viscosity improving the tightness of the cement matrix. However, the addition of nanosilica, whose particles are much finer than microsilica, causes a rapid decrease in the consistency of the fresh mix of cement mortars. The microsilica and nanosilica were often used together with other waste materials, i.e., broken glass [8,9,10,11,12], waste ceramics [1], or fly ash [5,13,14,19]. The use of waste is particularly important if we want building materials to be produced in accordance with the idea of a circular economy.

Various approaches have been explored to enhance the mechanical properties of cementitious materials, including the incorporation of additives, that can modify the microstructure and hydration characteristics of cement-based systems, thereby influencing their mechanical behavior. A new trend in research is the use of not only microsilica, which is a waste from the metallurgical industry (created during the processing of metallic silicon, ferrosilicon, or other silicon alloys), but also the use of biosilica of biogenic origin or extracted from rice husk ash using the thermochemical method [22,23,24,25,26,27,28]. Using diatomite (DE) as a reinforcement in cement-based materials has shown promising results [25]. These materials exhibit higher resistance to leaching, improved Cl impermeability, and reduced drying shrinkage, making them suitable for use in reinforced structures located in freshwater, seawater, or humid environments. Alternatively, DE-containing cement-based composites can serve non-reinforced structures (e.g., concrete blocks) [26,27]. It is well known that biosilica, a naturally occurring silica derived from diatoms, has been proposed as a potential additive to improve the mechanical properties of cement-based materials. Biosilica has several advantages over traditional silica-based additives, including its biodegradability and renewable nature [22]. The application of biosilica as an additive in cement mortar holds significant importance in the field of construction materials. Its unique properties, including high surface area, pozzolanic activity, and micro-sized particles, present a promising avenue to address the challenges of improving the mechanical properties of cement-based materials.

In a recent study [28], it was demonstrated that DE with a biosilica content of 83+ wt.% can effectively replace up to 40 wt.% of Portland cement in mortar and concrete, resulting in significant improvements in compressive strength. By partially replacing Portland cement with high-quality DE, the study found that the global warming potential could be reduced by 50% and emissions of PM10 and lead from mortar/concrete production could also be minimized, in addition to the strength enhancements [29]. In [22], it was shown that the compression strength of biocement mortar was greatly enhanced with the addition of biosilica. Nevertheless, the inclusion of biosilica particles resulted in a reduction in both the water vapor transmission rate and the setting time of the mortar. In [23], the researchers investigated the durability performance and underlying methods of DE-containing cementitious composites by partially replacing Portland cement with 30 wt.% diatomite. The results showed that the cementitious materials containing DE biosilica exhibited remarkable impermeability and leaching resistance after 28 days.

The effectiveness of an additive in cement mortar depends not only on its inherent properties but also on the method of incorporation into the mortar mix. Mixing methods play a crucial role in achieving a homogeneous dispersion of additives, ensuring their proper interaction with the cementitious matrix, and influencing the overall performance of the resulting composite. Therefore, understanding the effect of mixing methods on the utilization of biosilica in cement mortar is vital for optimizing its effectiveness and harnessing its full potential.

With the aim of contributing to the advancement of construction materials, this research delves into the intriguing application of biosilica in cement mortar, emphasizing the significant impact of mixing techniques on the resultant compressive strength. In this work, the effect of biosilica concentration and two different mixing methods with Portland cement on the compressive strength of cement-based mortar were investigated.

## 2. Materials and Methods

### 2.1. Materials

As a binder for the mortars in this study, ordinary Portland cement 52.5 (GOST 31108-2020) was used, which is available at Ararat cement Factory in Yerevan, Armenia. Table 1 displays the chemical composition and physical properties of the cement used in the study [30,31], in accordance with GOST EN 196-1-2002 [32], 196-2-2002 [33], and 196-3-2002 [34]. Meanwhile, Table 2 and Table 3 provide the physical and chemical properties of the sand and biosilica, respectively, that were used in the study. The biosilica (Effect group, Yerevan, Armenia) used as an additive in these mortars, which is amorphous silica of biogenic origin that resulted in the combined activation of the natural diatomite. The FTIR spectra and SEM image of the above-mentioned biosilica have been presented in Figure 1a,b, respectively. Several characteristic absorption bands can be observed in the spectra, providing insights into the molecular structure of biosilica. In the FTIR spectrum the general peak observed was around 1050–1100 cm^−1^, which corresponds to the characteristic Si-O-Si stretching vibrations. This peak confirms the presence of silica (SiO_2_) in the biosilica sample, which is essential for its potential pozzolanic activity and contribution to the mechanical properties of cement mortar. On the other hand, from the SEM image, it can be observed that the biosilica particles exhibit a predominantly spherical or irregular shape. As can be seen from Figure 2, the particle size distribution (ranging from nanoscale to micrometer scale) appears to be relatively uniform, which suggests that the biosilica promotes its potential interaction with the cementitious matrix.

### 2.2. Mixing and Sample Preparation

The w/c ratio used in the present study was 0.47, and the used cement/sand proportion was 1:4 [31]. Two different mixing methods of biosilica were used for sample preparation. First, biosilica and cement were mixed (E095 Mortar mixer, Matest, Treviolo, Italy) for 2.0 min, then this mixture and sand were mixed for 2 min, and the final mixture was mixed with water for 5 min. In the case of second mixing method, the cement and sand were mixed for 2 min, biosilica and water were mixed with magnetic mixer for 5 min, then both obtained mixtures were mixed together for 5 min. The size of the molds were 40 mm × 40 mm × 160 mm. Using a vibrating table (C278, Matest, Treviolo, Italy) the mortar was subjected to 30 s of compaction. Likewise, various mortars were created, including one with no biosilica and others with biosilica content of 2.5, 5.0, 7.5, and 10% of the weight of the cement. After 24 h, the specimens were de-molded, and the mortar samples was immersed in water at 20 ± 0.2 °C (Figure 3).

### 2.3. Compressive Strength and Water Absorption Testing

Three specimens were randomly selected from each batch to measure their compressive strength using the Concrete Compression Machine 2000 kN automatic, Servo-Plus Progress, in accordance with standard EN 196-1. The measured specimen dimensions for testing compressive strength were 40 mm × 40 mm. The compressive tests were performed at the ages of 7 and 28 days using an automatic pressure machine (C089) with a loading rate of 2.4 kN/s. In this study, we also paid attention to water absorption characteristics, with tests being conducted in accordance with GOST 12730.3-2020 [35].

Water absorption refers to the capacity of a material to absorb and retain water, and it is assessed by measuring the water saturation of a test sample [GOST 12730.3-2020 Concretes. Method of determination of water absorption]. The test samples were dried at 105 °C until they reached a constant weight and were then weighed under air-dry conditions (*m*_1_). Then, to determine the mass of the saturated test samples, they were submerged in a container of water at a temperature of (20 ± 2) °C, ensuring that the water level was positioned 50 mm above the upper mark of the test samples. At 24 h intervals, the test samples were weighed under air conditions with an accuracy of no more than 0.1% (*m*_2_). Before weighing, the surfaces of the saturated samples were wiped with a damp cloth. The test sample is considered saturated when the difference between consecutive weighings does not exceed 0.1%. After completing the aforementioned steps, the water absorption (W) of the test samples was determined using the following formula:$$W=\frac{m_{2}-m_{1}}{m_{1}}\;\cdot \;100\%\;$$
where W is the water absorption in percentage, *m*_2_ is the mass of the test sample after saturation, and *m*_1_ is the mass of the air-dried test sample. By applying this formula, the water absorption value can be calculated based on the weight difference between the saturated and air-dried samples, expressed as a percentage of the initial dry weight.

## 3. Results

Figure 4a,b shows the compressive strength of cement mortars with different wt.% of biosilica for 7 and 28 days, respectively. B0, B1, B2, B3, and B4 correspond to the 0%, 2.5%, 5%, 7.5%, and 10% of biosilica, respectively. The presented results were obtained when biosilica was mixed with cement (mixing method 1). The results indicate that the compressive strength of each composition increases with the increase in curing period. This is associated with increased hydration over time. On the other hand, the compressive strength increases with the increase in biosilica concentration. It is shown that for all cases the compressive strength has the maximum value of 10% biosilica concentration. In particular, the strength of the specimens over 7 days of curing increased by 30.5%, and by 36.5% for a curing period of 28 days.

Figure 5a,b shows the compressive strength of cement mortars with a different wt.% of biosilica for 7 and 28 days, respectively. These results were obtained when biosilica was mixed with water (mixing method 2). The same effects of the above parameters (curing period and biosilica concentration) on the compressive strength were also observed for this mixing method. In this case, the compressive strength of the specimens over 7 days of curing increased by 23.4%, and by 47.3% for a curing period of 28 days.

As can be seen from Figure 3 and Figure 4, in all cases, by increasing the content of biosilica in cement mortar, the porosity of the interfacial transition zone significantly diminishes, and that is why the compressive strength always increases (in the Appendix A we present the flexural strength results).

Figure 6a,b shows the water absorption of cement mortars with a different wt.% of biosilica for 28 days in the case of using two different mixing methods (method 1—Figure 5a, method 2—Figure 5b), for which the parameters are already presented above. From the obtained results, the minimum value of water absorption has the same 10% biosilica concentration. In the case of mixing method 1, the water absorption parameters reduced by 22%, and 34% in the case of mixing method 2.

The incorporation of a substantial amount of biosilica can result in the filling of gaps and consequently enhance the compressive strength and reduce water absorption. Biosilica particles can absorb compressive loads and thus provide high strength. Due to their small size, biosilica particles can settle into the voids in cement, leading to increased density. The pores become nondeformable due to the presence of biosilica particles, which helps maintain structural integrity. It follows that a homogeneous distribution of biosilica particles can underlie the results obtained due to various mixing methods, and it is obvious that the degree of particle distribution homogeneity is higher in the case of the second mixing method.

## 4. Conclusions

In this work, the mechanical properties such as compressive strength and water absorption of cement-based mortar with different concentrations of biosilica have been investigated. The mortar was prepared using the following two biosilica mixing methods: First, biosilica was mixed with cement and appropriate samples were prepared. For the other mixing method, samples were prepared by dissolving biosilica in water using a magnetic stirrer. It is shown that, for all cases, the compressive strength has the maximum and water absorption has the minimum value for 10% biosilica concentration. The obtained results show that, in the case of first mixing method, the compressive strength of the specimen over 7 days of hardening increased by 30.5%, and by 36.5% for a curing period of 28 days. In the case of the second mixing method, the compressive strength of the specimen over 7 days of hardening increased by 23.4%, and by 47.3% for a curing period of 28 days. Using mixing methods 1 and 2, the water absorption parameters reduced by 22% and 34%, respectively. 

It can also be concluded that using mixing method 2 allows for an increase over 28 days in the mortar’s compressive strength by 29.6% and a reduction in water absorption by 35.3% compared with mixing method 1.

Finally, we can conclude that different mixing methods will lead to variations in the dispersion and distribution of biosilica within the mortar matrix, thereby affecting the resulting compressive strength and water absorption. By systematically studying and comparing different mixing methods, this study intends to provide valuable insights into the optimization of biosilica incorporation in cement mortar.

## Figures and Tables

**Figure 1 materials-16-05516-f001:**
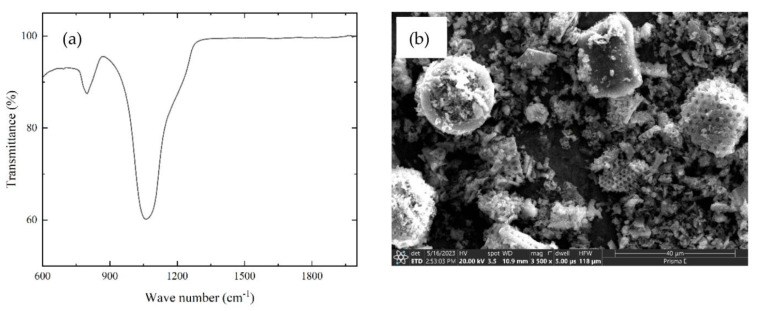
FTIR spectra (**a**) and SEM image (**b**) of biosilica.

**Figure 2 materials-16-05516-f002:**
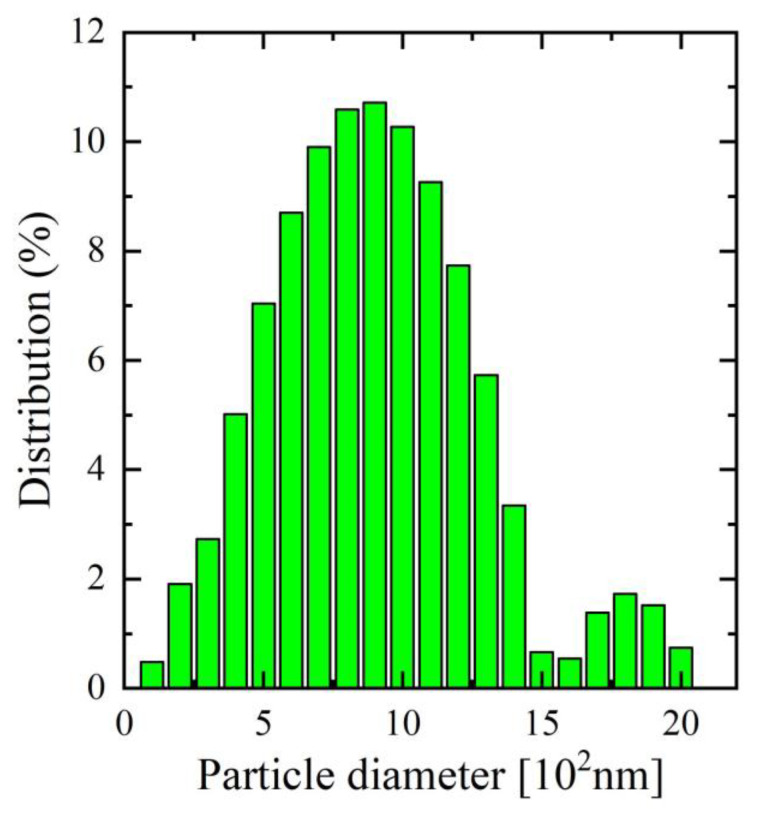
Particle size distribution of biosilica.

**Figure 3 materials-16-05516-f003:**
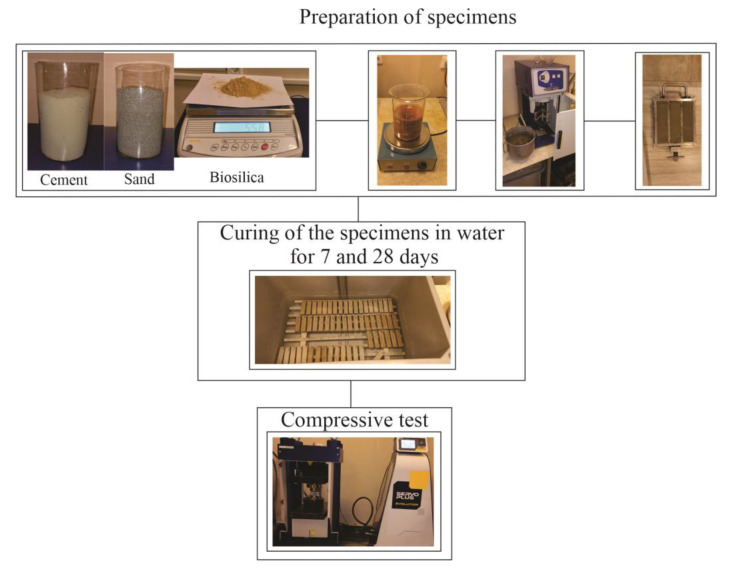
Diagram of the experimental procedure.

**Figure 4 materials-16-05516-f004:**
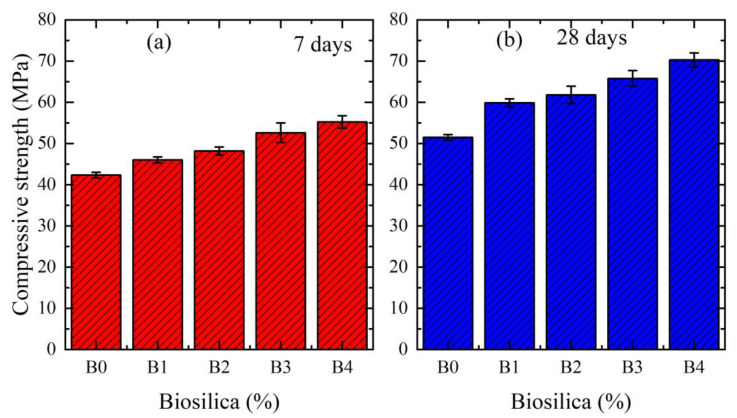
Compressive strength of cement mortars with a different wt.% of biosilica. The results are given for mixing method 1. (**a**) for 7 days, (**b**) for 28 days.

**Figure 5 materials-16-05516-f005:**
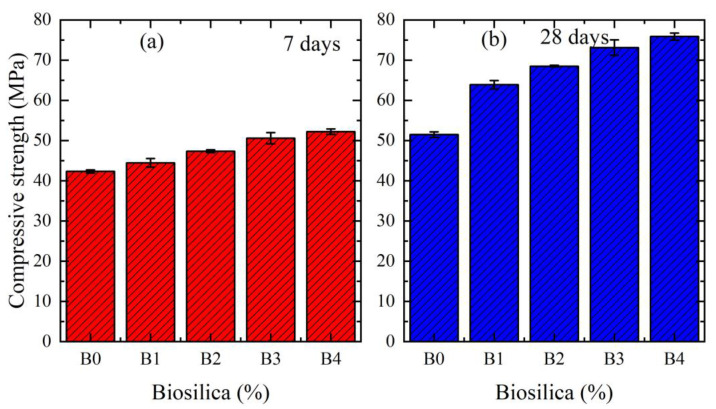
Compressive strength of cement mortars with a different wt.% of biosilica. The results are given for mixing method 2. (**a**) for 7 days, (**b**) for 28 days.

**Figure 6 materials-16-05516-f006:**
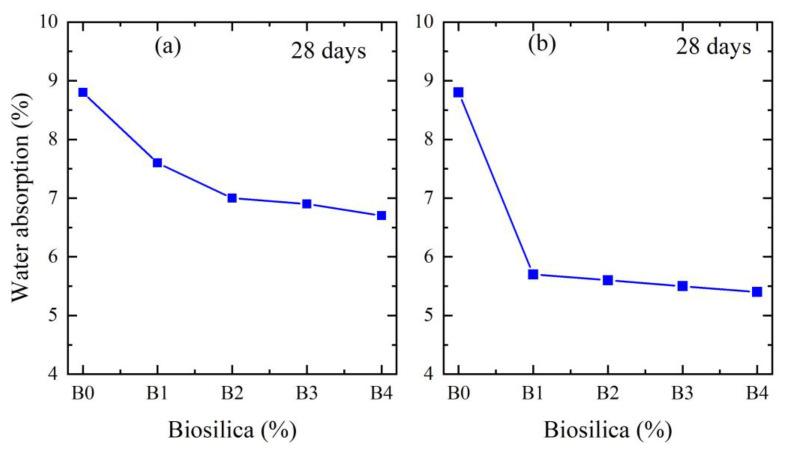
Water absorption of cement mortars with a different wt.% of biosilica. The results are given for mixing method 1 (**a**) and mixing method 2 (**b**), respectively.

**Table 1 materials-16-05516-t001:** Physical properties and chemical composition of cement.

Characteristics			Days			Results Obtained		
Standard consistency (%)			-			30		
Specific gravity (g/cm^3^)			-			3.1		
Blain’s fineness (m^2^/kg)			-			354.8		
Compressive strength (MPa)			3 days			21		
			7 days			37		
			28 days			51		
Setting time (min)			Initial			50		
			Final			310		
Chemical composition of cement (wt.%)								
Al_2_O_3_	SiO_2_	Fe_2_O_3_	CaO	MgO	SO_3_	Loss of ignition	Insol. Resid.	Free CaO
3.93	21.9	2.17	62.2	1.1	2.1	3.2	1.9	1.5

**Table 2 materials-16-05516-t002:** Physical properties of sand.

Fineness Modulus	Specific Gravity	Zone	Bulk Density in Compact State (kg/m^3^)	Bulk Density in Loose State (g/cm^3^)
2.35	2.44	II	1739	1.57

**Table 3 materials-16-05516-t003:** The metal oxide content in biosilica (wt.%).

SiO_2_	Al_2_O_3_	Fe_2_O_3_	K_2_O	MgO
88.92	6.1	2.8	1.34	0.84

## Data Availability

The data presented in this study are available upon request from the corresponding author.

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
