# Peer review of "The Use of Biosilica to Increase the Compressive Strength of Cement Mortar: The Effect of the Mixing Method"

_materials, 2023, doi:10.3390/ma16165516_

Round 1

Reviewer 1 Report

This paper presented a laboratory research on the effect of biosilica on compressive strength and water absorption of cement mortar in two mixing methods comparatively. The tests are well presented and conclusions sufficiently supported. However, the paper can be further improved in several ways:

1. The Introduction doesn't echo with your research well, in other words, your work doesn't solve the problems/deficiency of the biosilica added to cement based materials shown in the Introduction. 

2. The purpose or the highlight is not clear. One major purpose is to compare different mixing procedures of adding biosilica. Although two mixing methods were employed and comparatively studied, this part should be elaborated in the analysis, highlighted in the conclusion and reflected in the title. 

3. The scope of the research is a bit narrow, there are many properties characterizing cement mortar in different aspects. Compressive strength and water absorption are two of them, but there are many more, i.e. flexural strength, permeability, shrinkage and so on. I suggest the authors re-submit the manuscript after conducting supplemental tests and analysis. 

The English writing is generally good, with a few minor errors, please double check. 

Reviewer 2 Report

Dear authors,

The topic touched on in this manuscript represents a considerable scientific interest as well as touches on important aspects of climate change concerns and the circular economy. 

Regarding the structure and content of the manuscript, there are several comments and suggestions for the authors to be considered and addressed.

1. Based on the provided literature review, the effect of DE biosilica on the mechanical performance of PC mortars has been studied by a number of researchers. From the text of the manuscript, it is not clear, what is the novelty of the current study and how it contributes to the already existing knowledge in the field. 

2. In regard to other sources of natural and sustainable nanosilica, some authors also explored the effect of nanosilica derived from hydrothermal solutions on the performance of PC-based materials. I would recommend mentioning the following research study in the introduction section of the manuscript: Flores-Vivian, I., Pradoto, R.G.K., Moini, M. et al. The effect of SiO2 nanoparticles derived from hydrothermal solutions on the performance of portland cement based materials. Front. Struct. Civ. Eng. 11, 436–445 (2017). https://doi.org/10.1007/s11709-017-0438-2.

3. Please, provide PSD data of the biosilica powder to support the particle size range of the product.

4. Please, provide SEM images of the mortars with biosilica at the age of 28 days to support statements on the densification of the PC matrix with the addition of nanosized silica.

5. Please, clarify if the sand used in the study was standard and provide an explanation of the Zone II parameter for sand presented in Table 2.

6.  Review the text of the manuscript for typos, especially in the yellow marked text, fix superscript formatting where necessary, and fix decimal separators (Table 2).

Minor formatting and grammar edits

Round 2

Reviewer 1 Report

The concerns have been properly addressed by the authors, I have no more comments. One last piece of advice is that, given the flexural test was conducted and provided as supplemental material, have the authors considered incorporating the results into the main body of the paper?

Author Response

Response to Reviewer 1

Dear Reviewer

Thank you very much for your question. The answer is included in the second revised version of the manuscript (used the yellow color).

Faithfully yours,

Dr. Manuk Barseghyan (on behalf of all authors)

National University of Architecture and Construction of Armenia, Armenia

Reviewer 2 Report

The manuscript can be recommended for further process of publication in its current state

Author Response

Response to Reviewer 2

Dear Reviewer

Thank you very much for your recommendation.

Faithfully yours,

Dr. Manuk Barseghyan (on behalf of all authors)

National University of Architecture and Construction of Armenia, Armenia
